# Do Gender Role Stereotypes and Patriarchal Culture Affect Nursing Students’ Major Satisfaction?

**DOI:** 10.3390/ijerph18052607

**Published:** 2021-03-05

**Authors:** Sunhee Cho, Sun Joo Jang

**Affiliations:** 1Department of Nursing, Mokpo National University, Muan 58554, Korea; scho@mokpo.ac.kr; 2Red Cross College of Nursing, Chung-Ang University, Seoul 06974, Korea

**Keywords:** gender role stereotype, major satisfaction, patriarchalism

## Abstract

This study aimed to identify the relationships between gender role stereotypes (GRS), patriarchal family environment, and major satisfaction (MS) and their associated factors in nursing students. A total of 195 nursing students (154 women, 41 men) were surveyed online in South Korea from May to June 2020. The Gender Role Stereotype Inventory was used to assess gender role stereotypes, while patriarchal family environment and MS were evaluated using a 11-item instrument for testing patriarchal family environment and the Major Satisfaction Inventory, respectively. Men demonstrated stronger gender role stereotypes and patriarchal family culture than women. Patriarchal family culture was significantly correlated with gender role stereotypes and MS. A multiple regression analysis was performed to identify the effects of age, academic performance, motive for MS, intellectual GRS, social GRS, and patriarchal family environment on MS. The explanatory power of this model was 12.2%. Younger age, higher grade point average, voluntary selection of major, lower intellectual gender stereotypes, and higher social gender stereotypes predicted higher MS. Further studies on cultures in Asia and the world are needed to understand the impact of the patriarchal family culture and gender role stereotypes of nursing students on school adaptation.

## 1. Introduction

As a sector, nursing consists mostly of women; thus, it is representative of a woman-dominant sector [1]. Traditionally, nursing has been considered a woman’s occupation, and the old stereotypes that the nature of nursing is feminine and is not suitable for the disposition of men have existed since the days of Nightingale [2]. Since the 2000s, the proportion of female college students studying nursing has been significantly high; however, changes in the job structure and an increased awareness on nursing as a profession have resulted in more male students applying to study nursing in university [3]. For example, 10,473 of the 52,256 nursing students in South Korea enrolled in 2019 were men, exceeding 20% of the total population [4]. This number has increased more than ten-fold, which is notable compared to the 1.8% increase observed in 2000. It is also greater than the 9.7% increase in the proportion of male nursing students from 2006 in the United States to the 11% increase in 2018. The significant growth in the number of male nursing students reflects the increased social awareness of nursing professions and simultaneously promotes changes in the university nursing education environment [5].

“Gender role conflict” denotes a psychological state that negatively affects an individual and others as a result of the individual’s excessive internalization of their expected gender role [6]. In gender role conflict theory, O’Neil [6] defined restrictions as an individual’s attempt to control their behavior and the behaviors of others to conform to stereotypical and restrictive norms, consistent with the ideology of masculinity. Gender role conflict appears when gender role stereotypes are strong. For our purposes, it is important to note that it tends to emerge in Korea in men who internalize the culture’s masculine values, such as “men must be strong” and “my problems must resolve themselves”. As part of female-dominant groups and, thus, a predominately feminine culture, Korean male nursing students find it difficult to overcome discomfort and alienation [7].

It is also helpful to take a moment here to establish the meaning of patriarchy. “Patriarchy” is a type of male-centered society in which men have power and play a monopolistic role in political leadership, moral authority, social privileges, and control over property. A patriarchal family environment with a clear distinction of gender roles can significantly impact the development of individual perceptions of gender roles [8]. Due to the influence of Confucian ideology, patriarchal values are deeply embedded in Korean society [9] and, although this has an important effect on various areas of life, few studies have been done on the effects of patriarchal values on nursing students.

Previous studies have shown that male nursing students and nurses perceived the existence of gender role stereotypes in Western cultures, such as the United States and the United Kingdom, as well as in Asian cultures with strong Confucian influence, such as Korea, China, and Japan [1,10]. Gender role stereotypes refer to the idea that certain roles must be performed by certain genders [11]. In general, traits such as domination, control, assertiveness, and aggression are perceived as masculine, and relation-oriented and dependent traits are regarded as feminine. Nightingale claimed that the nature of nursing is feminine, and nursing has been stereotyped as a female-dominated profession [2]. To date, nursing is still considered a role for women, and there is a gender bias that puts male nursing college students and nurses in a position that encourages gender identity issues [5]. A recent study in Korea showed a negative correlation between gender role stereotypes and how satisfied male nursing students were with their major [12]. Further restriction of such gender role stereotypes can lead to conflicts in gender roles with negative consequences [13].

In the era of globalization, culture is an essential concept in nursing [14]. Traditionally, Korea has been under the influence of Confucianism with moral and philosophical rather than religious meanings [9]. Korea is one of the main patriarchal countries in the world [14], with a cultural atmosphere that distinguishes men and women from each other and favors men in families. This patriarchal social structure has an important influence on determining the role of gender [15], and those raised in a patriarchal family environment have a strong stereotypical idea of gender roles. More specifically, they do not believe in gender equality [16], and they believe that women and men have separate roles [17]. Coming of age in this patriarchal culture, which requires passive, dedicated nurses within hospitals and considers nursing a feminine role [1], the emerging generation of women, which values gender equality and autonomy, is expressing dissatisfaction with these gendered social perceptions of nurses [18]. Another study reported that patriarchal gender role stereotypes affect gendered vocational aspirations [19]. Based on this evidence, it is believed that patriarchal culture is closely related to gender role stereotypes, which are thought to affect both male and female nursing students. Therefore, this study aimed to investigate the association among gender role stereotypes, patriarchal culture, and the major satisfaction (MS) of male and female nursing students in Korea.

## 2. Materials and Methods

### 2.1. Study Design, Participants, and Ethical Considerations

This was a cross-sectional study that assessed the relationship between the main variables, gender role stereotypes, patriarchal family environment, and MS in nursing college students, and identified factors that affect MS. Announcements were posted in three university campuses in Korea calling on second- to fourth-year nursing students who were willing to participate in the survey. First-year students were excluded because they had not yet enrolled in major courses. Students who wanted to participate voluntarily provided their informed consent (i.e., they checked a box in our form) and completed the self-administered questionnaire electronically distributed by a hyperlink. The completed questionnaires were collected anonymously between May and June 2020. The minimum sample size required for multiple linear regression analysis was calculated using the G*power 3.1.9.2 Version program with a significance level of 0.05, power of 0.90, 9 predictor variables, and an effect size of f = 0.15 (medium). The calculated sample size was 166, and 208 participants were set as the intended sample size considering a 20% drop-out rate. Data were initially collected from 231 participants; however, answers repeated by the same participants were excluded, bringing the final number of participants to 195.

Data collected online did not include information that could be used to identify the subjects. Additionally, they were directly coded and stored, guaranteeing the autonomy and anonymity of the participants. This study was conducted after obtaining approval from the Institutional Review Board of Mokpo National University (MNUIRB-200617-SB-006-01).

### 2.2. Measures

#### 2.2.1. Gender Role Stereotype

The gender role stereotype was assessed using the 5-point Likert scale of the Gender Role Stereotype Inventory [11], which was a tool developed to assess the complex concept of gender stereotypes. It is an 18-item instrument that consists of five subfactors, including familial factors (stereotypes of gender roles in the family), occupational and external factors (stereotypes of occupations and appearance characteristics of men and women), psychosocial factors (stereotypes of personality characters), intellectual factors (stereotypes of intellectual characteristics between men and women), and social factors (gender role stereotypes in social life). The participants were asked to rate each item on a scale of 1–5, with 1 indicating “strongly disagree” and 5 indicating “strongly agree”. The total gender role stereotype score was calculated by adding all the scores from each item, yielding values ranging from 18 to 90. A higher score indicated a stronger stereotype.

#### 2.2.2. Patriarchal Family Environment

We used an 11-item instrument for evaluating the patriarchal family environment [20] to measure the patriarchal attitudes of participants and their parents. The instrument is a 5-point Likert scale with a score of 1 indicating “strongly disagree” and 5 indicating “strongly agree”. The total patriarchal family environment score can be calculated by adding all the scores from each item, yielding values ranging from 11 to 55. A higher score indicates a more patriarchal family.

#### 2.2.3. Major Satisfaction

MS was assessed using the scale developed by Kim and Ha [21]. It measured participants’ satisfaction with curriculum content, interest in nursing, and social perception of nursing. The 13-item instrument is a 5-point Likert scale with a score of 1 indicating “strongly disagree” and 5 indicating “strongly agree”. The total MS score can be calculated by adding all the scores from each item, yielding values ranging from 13 to 65. A higher score indicates a higher level of satisfaction.

#### 2.2.4. Validity, Reliability, and Rigor

All instruments used in this study have been validated in previous research. Cronbach’s alpha scores for each instrument are as follows: gender role stereotypes = 0.87 [11] and 0.92 [12]; patriarchal family environment = 0.79 [20]; and MS = 0.89 [21] and 0.97 [12]. Six nursing professors with more than 10 years of educational experience and experience in developing tools verified the content validity. The scale content validity index (S-CVI)/averages for each instrument are as follows: gender role stereotypes = 0.99, patriarchal family environment = 0.99, and MS = 0.99. The S-CVI/universal for each instrument are as follows: gender role stereotypes = 0.94, patriarchal family environment = 0.91, and MS = 0.92.

### 2.3. Data Analysis

SPSS 26.0 (IBM, Armonk, New York, NY, USA) was used for data analysis. Descriptive statistics such as frequency, percentage, mean, and standard deviation were used to analyze the general characteristics of the participants. Differences in gender role stereotypes, patriarchal family environment, and MS according to the general characteristics of participants were identified using independent t-tests, analysis of variance (ANOVA), and Scheffé tests. Pearson’s correlation coefficient was used to analyze correlations between gender role stereotypes, patriarchal family environment, and MS. Finally, a multiple regression analysis using the enter method was conducted to identify the impact of gender role stereotypes, patriarchal family environment, and the general characteristics on MS. The Kolmogorov–Smirnov test was used to verify normality, and the Koenker test was used to verify the homoscedasticity of residuals for goodness-of-fit in the regression model.

## 3. Results

### 3.1. General Characteristics of Participants

The general characteristics of the participants are summarized in Table 1. Of the participants, 79.0% were women. The mean age was 22.79 years. Almost all of the participants (89.2%) had a middle socioeconomic status (SES). The grade point average (GPA) of the previous semester was 3.40 (out of 4.50). The majority of participants were fourth-year students (50.8%), followed by second- and third-year students. Of the participants, 27.7% chose to major in nursing as they were interested in it and believed it was appropriate, while the remaining 72.3% stated that others had advised them to major in nursing.

### 3.2. Gender Role Stereotypes, Patriarchal Family Environment, and MS According to Participants’ General Characteristics

As shown in Table 2, there was no difference in MS according to gender. However, male students had greater gender stereotypes (*t* = 3.58, *p* < 0.001) and indicated that they had been raised in more patriarchal family environments (*t* = 3.55, *p* < 0.001) than female students. The gender role stereotypes, patriarchal family environment, and MS according to year and SES were also not significantly different. Regarding the MS motive, MS was higher in those who voluntarily chose to major in nursing than those who were advised by others to major in it (F = −3.53, *p* = 0.001).

### 3.3. Correlation among Gender Role Stereotypes Subfactors, Patriarchal Family Environment, and MS

Occupational gender stereotypes (r = 0.148, *p* = 0.039) and social gender stereotypes (r = 0.182, *p* = 0.011) showed a weak positive correlation with age, according to Table 3. All subfactors of gender stereotypes were moderately correlated with patriarchal family environment, while MS showed a weak negative correlation with age and patriarchal family environment (r = 0.149, *p* = 0.037).

### 3.4. Factors Influencing MS

Prior to carrying out the multiple regression analyses, we evaluated autocorrelations of the dependent variables and the multicollinearity of the independent variables. The Durbin–Watson index confirmed that there was no autocorrelation in the dependent variable. For the regression model, we selected the predicting variables associated with MS. The variance inflation factor and tolerance limit indicated the absence of multicollinearity (Table 4). Further, regression results revealed that age (β = −0.18, *p* = 0.017), reasons for choosing the major (β = −0.22, *p* = 0.001), GPA (β = 0.16, *p* = 0.020), intellectual gender role stereotypes (GRS) (β = −0.19, *p* = 0.032), and social GRS (β = 0.16, *p* = 0.031) significantly predicted MS; this increased with lower age, voluntary selection of the major, lower intellectual GRS, and higher social GRS. However, gender and patriarchal family environment were not statistically significant predictors of MS. Through extra regression analysis, we found that gender role stereotypes did not play a mediating role in the relationship between patriarchal family environment and MS (B = −0.14, *p* = 0.079). These predictors accounted for 12.2% of the variance in MS. The goodness-of-fit of the regression model was tested with a residual analysis to verify the normality and homogeneity of variance. The Kolmogorov–Smirnov test results (Z = 0.06, *p* = 0.553) satisfied the normality assumption, and the Koenker test results satisfied the homoscedasticity of the variance assumption. Hence, the model was deemed to be a good fit for the data (Table 4).

## 4. Discussion

The purpose of this study was to assess gender role stereotypes and patriarchal family environment and their association with MS in nursing students. The main points of the discussion based on the findings of this study are as follows.

First, among demographic characteristics, there were differences in gender role stereotypes and patriarchal family environment according to gender. Male students showed greater gender stereotyping and were shown to have grown up in a more patriarchal family environment. This finding is consistent with that of a previous study in which males showed a patriarchal gender role orientation and a gender style typical of their vocational aspiration [16]. Similarly, this finding is also in line with the results of previous studies that showed that women tend to demonstrate a mindset geared more toward gender equality [1,15].

A previous study on nurses [1] showed that male nurses perceived managerial roles in nursing professions according to gender stereotypes; more specifically, they associated managerial roles more closely with men than women compared to female nurses. Moreover, in a previous study in Turkey and Korea, two countries with patriarchal cultures [14], male nursing students showed more sexist attitudes than female nursing students. Such gender stereotyping suggests the possibility of another form of gender segregation and that medical institutions should ensure gender equality in managerial roles. In particular, gender roles may be reversed in traditional patriarchal institutions such that employment, education, performance assessment, and promotion opportunities may only be in favor of men [1,22]. Meanwhile, another previous study [23] observed that many nursing students initially had their own gender role stereotypes of nursing professions, which changed over time with coursework, education, and training. In our study, there was no statistically significant difference in gender role stereotypes by year. Since the fourth-year students were the majority of participants and first-year students were excluded from the study, it was difficult to investigate gender role stereotypes by student grade. Therefore, future studies should identify differences in gender role stereotype according to grade by recruiting participants using stratified sampling that considers the proportions of students per class.

A moderate or higher correlation was observed between gender role stereotypes and patriarchal family environment, and there was a positive correlation between all their subfactors. This finding is consistent with that of a previous study [15], which found that ideologies of gender equality were associated with nonpatriarchal family backgrounds. Moreover, our study’s multiple regression analysis showed that patriarchal family environment did not affect MS but only weakly negatively correlated with it. In addition, gender role stereotypes were not correlated with MS. However, the multiple regression analysis showed that intellectual GRS and social GRS impacted MS in nursing students.

Gender role stereotypes are rooted in family [19,24], and patriarchy-induced distorted gender role schemata can promote gender role conflict [6], negatively affecting MS. However, our findings suggest that the influence of a patriarchal cultural background can be overcome by demographic characteristics, motives for MS, and academic achievement. This is also supported by the results of previous studies [25,26,27], which showed that motive can affect MS. In these studies, MS was highest in students who voluntarily chose their majors based on suitability and interest.

Among the subfactors of GRS, only intellectual GRS and social GRS affected MS; meanwhile, familial, occupational, external, and psychosocial GRS did not significantly affect MS. Indeed, fewer intellectual stereotypes about gender, such as “men are more creative than women” and “men are more rational than women”, predicted higher MS in nursing students. This result is consistent with our expectations and with most findings from a previous study on gender stereotypes [12]. However, contrary to our expectations, we found that MS was higher when gendered stereotypes of social roles existed.

This is similar to the findings from a previous study on nursing students [28], which reported that differences in biological sex, as well as those in gender role orientation, did not affect critical thinking and caring behaviors. Moreover, high levels of femininity and masculinity resulted in a more caring behavior, which increased the level of critical thinking in students. In a qualitative study on gender role stereotypes, Rabie et al. [29] argued that gender cannot predict nursing ability, and a previous study warned that incorrect norms and perceptions of gender roles in the nursing profession can be harmful [13]. It is believed that students who voluntarily chose to major in nursing use an interpretative framework that affirms gendered sociocultural norms, resulting in differences in gender roles and biological differences in gender. In other words, the findings of our study suggest that our participants were aware of the inherent strengths and weaknesses of each gender, rather than being antagonistic with others of the same and different genders. Additionally, the participants demonstrated realistic expectations of their gender roles, which reflects their maturity in the aspect of gender role conflict.

In the past, nursing failed to normalize feminism’s understanding that men and women should be treated as equals. Feminist ideology was only accepted and applied after feminism, which recognizes and promotes the unique characteristics of each gender, was established in nursing [30]. Along these lines, gender stereotypes in nursing may need to be viewed from an integral perspective. As Evans [22] suggested, the roles of men and women in nursing need to be gradually merged, rather than consider nursing as a quintessential feminine aspect and dividing gender roles.

Because this study only included nursing students from three universities in South Korea, there is a possibility of selection bias; therefore, the results have limited generalizability. Additionally, the possibility that male students with high gender role stereotypes or patriarchal family environments did not choose nursing as their college major cannot be negated. Subsequent studies would do well to verify the difference in gender role stereotypes and patriarchal family environment between male students who choose nursing majors and male students from other majors. Moreover, the cross-sectional design of this study limited the evaluation of the effects of patriarchal family environment and gender role stereotypes on MS. To make this line of research more comprehensive, we encourage future scholars to perform comparative studies of nursing students from various cultures around the world. Additionally, we also advise that scholars carry out a large-scale prospective cohort study on nursing college students to identify role conflicts in nursing tasks after the students become professional nurses.

In spite of the above limitations, this study offers significant useful insights for better understanding the situation of nursing college students by linking gender role stereotypes and satisfaction with nursing major to the patriarchal family environment characteristic of Asian cultures. In addition, this study is meaningful in that it has laid the foundation for faculty to understand and mediate the gender role stereotypes of nursing students by understanding that these stereotypes of nursing students can positively influence MS.

## 5. Conclusions

This study was conducted to understand the relationship between gender role stereotypes, patriarchal family environment, and MS. Age, academic performance, motive for MS, intellectual gender role stereotypes, and social gender role stereotypes had significant associations with MS, while patriarchal family environment did not. This model had an explanatory power of 12.2% and indicated that being younger in age, having a higher GPA, voluntarily selecting the nursing major, and having lower intellectual gender stereotypes and higher social gender stereotypes were all associated with increased MS. Overall, neither gender stereotypes nor patriarchal cultural background had a negative relationship with the major satisfaction of nursing students in this study. These findings imply that to increase MS and reduce the attrition rate in nursing students, nursing professionals and faculty members should work to understand their gender role stereotypes and patriarchal family environments. Additionally, gender neutrality and integrated gender role identification are required for nursing professionals.

## Figures and Tables

**Table 1 ijerph-18-02607-t001:** General characteristics of participants (*n* = 195).

Variable	Category	*n* (%)	M (SD)
Gender	Male	41 (21.0)	
	Female	154 (79.0)	
Age (year)			22.79 (1.63)
GPA			3.40 (1.02)
Grade	2	53 (27.2)	
	3	43 (22.1)	
	4	99 (50.8)	
SES	Low	20 (10.3)	
	Middle	174 (89.2)	
	High	1 (0.5)	
Motive for major selection	Voluntary decision	54 (27.7)	
	Recommendation from others	141 (72.3)	

**Table 2 ijerph-18-02607-t002:** Gender role stereotypes, patriarchal family environment, and major satisfaction (MS) according to participants’ general characteristics (*n* = 195).

Variable	Category	GRS		PFE		MS	
		M (SD)	t/F (*p*)	M (SD)	t/F (*p*)	M (SD)	t/F (*p*)
Gender	Male	37.73 (10.26)	3.58 (<0.001)	19.76 (7.27)	3.55 (<0.001)	47.17 (8.35)	0.05 (0.964)
	Female	31.87 (9.06)		16.08 (5.47)		47.23 (6.65)	
Grade	2	33.94 (9.14)	0.53 (0.576)	17.81 (6.11)	1.14 (0.323)	47.55 (6.05)	0.62 (0.537)
	3	33.70 (9.54)		15.98 (4.93)		48.02 (7.60)	
	4	32.39 (9.90)		16.72 (6.46)		46.69 (7.25)	
SES	Low	33.30 (11.00)	2.20 (0.114)	17.35 (6.00)	0.14 (0.869)	45.15 (12.67)	1.27 (0.282)
	Middle	32.97 (9.37)		16.78 (6.10)		47.48 (6.07)	
	High	53.00		19.00		42.00	
Motive for major selection	Voluntary decision	32.52 (9.69)	0.53 (0.600)	15.91 (5.30)	1.35 (0.179)	50.00 (5.03)	3.53 (0.001)
	Recommendation from others	33.33 (9.59)		17.21 (6.31)		46.15 (7.38)	

**Table 3 ijerph-18-02607-t003:** Correlation between general characteristics, gender role stereotypes, patriarchal family environment, and MS (*n* = 195).

					GRS					PFE	MS
		Age	GPA	Family	Occupation	Psychosocial	Intellectual	Social	Overall GRS		
Age		1									
GPA		−0.121	1								
G R S	Family	0.096	0.031	1							
	Occupation	0.148 *	0.099	0.693 ***	1						
	Psychosocial	0.069	−0.037	0.383 ***	0.468 ***	1					
	Intellectual	−0.033	−0.013	0.461 ***	0.499 ***	0.512 ***	1				
	Social	0.182 *	0.012	0.266 ***	0.296 ***	0.154 *	0.271 **	1			
	Overall GRS	0.129	0.021	0.858 ***	0.809 ***	0.700 ***	0.698 ***	0.494 ***	1		
PFE		0.123	−0.071	0.464 ***	0.542 ***	0.445 ***	0.311 ***	0.252 ***	0.566 ***	1	
MS		−0.149 *	0.202 **	−0.053	−0.071	−0.052	−0.127	0.081	−0.058	−0.149 *	1

* *p* < 0.05, ** *p* < 0.01, *** *p* < 0.001.

**Table 4 ijerph-18-02607-t004:** Factors influencing major satisfaction (*n* = 195).

Independent Variable	B	SE	β	t	*p*	VIF	95% CI	
							Lower	Upper
(Constant)	64.18	7.73		8.30	<0.001 ***		48.93	79.43
Age	−0.76	0.32	−0.18	−2.40	0.017 *	1.20	−1.39	−0.14
Gender ^†^	1.55	1.30	0.09	1.20	0.234	1.27	−1.01	4.12
Reasons for choosing the major ^††^	−3.47	1.06	−0.22	−3.27	0.001 **	1.02	−5.56	−1.37
GPA	1.12	0.48	0.16	2.34	0.020 *	1.07	0.18	2.07
GRS—intellectual	−0.82	0.38	−0.19	−2.16	0.032 *	1.67	−1.56	−0.07
GRS—social	0.54	0.25	0.16	2.18	0.031 *	1.20	0.05	1.02
Patriarchal family environment	−0.17	0.10	−0.15	−1.72	0.087	1.60	−0.37	0.03
Adj R^2^ = 12.2%								
F = 3.69, *p* < 0.001								

Durbin–Watson index = 1.97, Durbin–Watson’s du (lower critical limit) = 1.87, 4-du (upper critical limit) = 2.13, Koenker test χ^2^ = 2.68 (*p* = 0.443), ^†^ Reference: Male, ^††^ Reference: Voluntary selection of major, * *p* < 0.05, ** *p* < 0.01, *** *p* < 0.001.

## Data Availability

The data presented in this study are available on request from the corresponding author and with permission of the Institutional Review Board of Mokpo National University.

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
