# Peer review of "Do Gender Role Stereotypes and Patriarchal Culture Affect Nursing Students’ Major Satisfaction?"

_ijerph, 2021, doi:10.3390/ijerph18052607_

Round 1

Reviewer 1 Report

This article reports on an investigation into the relationship between gender role stereotypes, patriarchal family environment, and nursing students satisfaction with their study major (with control for GPA, and age). It was conducted in three universities in South Korea. This is an interesting and contemporary question to investigate, that from a methodological perspective, appears to have been rigorously and well analyzed in this study. I was excited to read this article and liked the bold approach taken by the authors. However, unfortunately, the article doesn't clearly present the underlying theory or the findings. Here I outline my main concerns.

  • The paper requires additional editing for the correct use of English. In particular, there are occasional instances of questionable word use (for example, the use egalitarian (a very broad term) to refer to gender equal mindsets), many instances of incorrect preposition use, some awkward switching between sex and gender (I would recommend the consistent use of gender), and some spelling mistakes (for example, felinity (L223) should be femininity).
  • Some inconsistent use of acronyms, and in general, an overuse of acronyms that detracts from the readability of the article. I would suggest not using acronyms for the two main concepts under investigation (GRS and PFE) and retaining the use of GRS only when discussing the sub-scales.
  • There is a liberal use of causal language (for example, affect, led to) that is not warranted by this type of study. It should be replaced with language that talks about observed relationships and/or associations.
  • The introduction is inadequate and does not fully position this study within the current literature. In addition, there is new theory introduced in the discussion section that should be moved into the introduction (the discussion section should only discuss the findings, and if reference to literature is made, it should be to the literature presented in the introduction). The introduction / literature section should be expanded, and should progress from general, to specific, to very specific to the study, to the research gap, to the research questions for this study, in a logical and systematic way. Such a systematic presentation of the body of knowledge that underpins this study is not evident.
  • Total GRS and the GRS sub-scales are presented and discussed, but only the subscales are presented in the tables. Some clarity is required as to why the GRS is only considered in its sub-scales and not as a total GRS score (as presented in lines 98-100).
  • Table 3 requires the heading for GRS to be reformatted, so it doesn’t appear to include age and GPA.
  • Final model – It appears that the final model includes all variables, although only the statistically significant variables are discussed. What would a parsimonious model look like (including only the statistically significant variables)? What would the goodness of fit statistics be on a parsimonious model? I feel that reducing the model to only the elements that have been found to be statistically significant, could then anchor a more ordered and clinical discussion of the results.
  • The discussion and conclusion do not make clear why this study, finally, is important, what it contributes to the body of literature, and how it can be useful to nursing training policies, programs and practitioners.

For these reasons, this article should be returned to the authors for major revisions, with an invitation for resubmission once the revisions are completed.

Author Response

Response to Reviewer 1

This article reports on an investigation into the relationship between gender role stereotypes, patriarchal family environment, and nursing students satisfaction with their study major (with control for GPA, and age). It was conducted in three universities in South Korea. This is an interesting and contemporary question to investigate, that from a methodological perspective, appears to have been rigorously and well analyzed in this study. I was excited to read this article and liked the bold approach taken by the authors. However, unfortunately, the article doesn't clearly present the underlying theory or the findings. Here I outline my main concerns.

  1. The paper requires additional editing for the correct use of English. In particular, there are occasional instances of questionable word use (for example, the use egalitarian (a very broad term) to refer to gender equal mindsets), many instances of incorrect preposition use, some awkward switching between sex and gender (I would recommend the consistent use of gender), and some spelling mistakes (for example, felinity (L223) should be femininity).

Response: We would like to thank you for your constructive suggestions and feedback. They have greatly improved the overall quality of our manuscript. Accordingly, we have revised as below.

(Page 6–7, Line 219-221)

“Similarly, this finding is also in line with the results of previous studies that showed that women tend to demonstrate a mindset geared more toward gender equality [1,15].”

- “Sex” was changed to “gender” (whole manuscript)

- Spelling mistakes were corrected (whole manuscript)

  1. Some inconsistent use of acronyms, and in general, an overuse of acronyms that detracts from the readability of the article. I would suggest not using acronyms for the two main concepts under investigation (GRS and PFE) and retaining the use of GRS only when discussing the sub-scales.

Response: Thank you very much. We have revised the whole manuscript accordingly.

  1. There is a liberal use of causal language (for example, affect, led to) that is not warranted by this type of study. It should be replaced with language that talks about observed relationships and/or associations.

Response: We would like to thank you for your constructive suggestions and feedback. They have greatly improved the overall quality of our manuscript. Accordingly, we have revised as below.

(Page 1, Line 30–32)

“…however, changes in the job structure and an increased awareness on nursing as a profession have resulted in more male students applying to study nursing in university [3].”

(Page 7, Line 262–264)

“Moreover, high levels of femininity and masculinity resulted in a more caring behavior, which increased the level of critical thinking in students. In a qualitative study on gender role stereotypes, …”

  1. The introduction is inadequate and does not fully position this study within the current literature. In addition, there is new theory introduced in the discussion section that should be moved into the introduction (the discussion section should only discuss the findings, and if reference to literature is made, it should be to the literature presented in the introduction). The introduction / literature section should be expanded, and should progress from general, to specific, to very specific to the study, to the research gap, to the research questions for this study, in a logical and systematic way. Such a systematic presentation of the body of knowledge that underpins this study is not evident.

Response: We would like to thank you for your constructive suggestions and feedback. They have greatly improved the overall quality of our manuscript. Accordingly, we have revised as below.

(Pages 1–2, Line 40–57)

““Gender role conflict” denotes a psychological state that negatively affects an individual and others as a result of the individual’s excessive internalization of their expected gender role [6]. In gender role conflict theory, O’Neil [6] defined restrictions as an individual’s attempt to control their behavior and the behaviors of others to conform to stereotypical and restrictive norms, consistent with the ideology of masculinity. Gender role conflict appears when gender role stereotypes are strong. For our purposes, it is important to note that it tends to emerge in Korea in men who internalize the culture’s masculine values, such as “men must be strong” and “my problems must resolve themselves.” As part of female-dominant groups and, thus, a predominately feminine culture, Korean male nursing students find it difficult to overcome discomfort and alienation [7].

It is also helpful to take a moment here to establish the meaning of patriarchy. “Patriarchy” is a type of male-centered society in which men have power and play a monopolistic role in political leadership, moral authority, social privileges, and control over property. A patriarchal family environment with a clear distinction of gender roles can significantly impact the development of individual perceptions of gender roles [8]. Due to the influence of Confucian ideology, patriarchal values are deeply embedded in Korean society [9] and, although this has an important effect on various areas of life, few studies have been done on the effects of patriarchal values on nursing students.”

  1. Total GRS and the GRS sub-scales are presented and discussed, but only the subscales are presented in the tables. Some clarity is required as to why the GRS is only considered in its sub-scales and not as a total GRS score (as presented in lines 98-100).

Response: We would like to thank you for your constructive suggestions and feedback. They have greatly improved the overall quality of our manuscript. Accordingly, we added the overall GRS score in Table 3. However, we are unable to present total GRS due to multi-collinearity; therefore, we only present subscale scores in Table 4.

  1. Table 3 requires the heading for GRS to be reformatted, so it doesn’t appear to include age and GPA.

Response: (Page 5, Line 186-187) As you recommended, we have revised the heading of Table 3 as below.

“Table 3. Correlation between general characteristics, gender role stereotypes, patriarchal family environment, and MS (N=195).”

  1. Final model – It appears that the final model includes all variables, although only the statistically significant variables are discussed. What would a parsimonious model look like (including only the statistically significant variables)? What would the goodness of fit statistics be on a parsimonious model? I feel that reducing the model to only the elements that have been found to be statistically significant, could then anchor a more ordered and clinical discussion of the results.

Response: Thank you for your helpful comments. For the parsimonious model, we included only the statistically significant variables as per your recommendation. As per the other reviewer’s advice, we used gender as the variable in the regression model.

We have revised Table 4 accordingly (see page 6, line 207).

  1. The discussion and conclusion do not make clear why this study, finally, is important, what it contributes to the body of literature, and how it can be useful to nursing training policies, programs and practitioners.

For these reasons, this article should be returned to the authors for major revisions, with an invitation for resubmission once the revisions are completed.

Response: Thank you for your helpful comments. Accordingly, we have revised as below.

(Page 8, Line 294–300)

In spite of the above limitations, this study offers significant insights useful for better understanding the situation of nursing college students by linking gender role stereotypes and satisfaction with nursing major to the patriarchal family environment characteristic of Asian cultures. In addition, this study is meaningful in that it has laid the foundation for faculty to understand and mediate the gender role stereotypes of nursing students by understanding that the social gender role stereotypes of nursing students can negatively influence MS.”

(Page 8, Line 310–314)

“These findings imply that to increase MS and reduce the attrition rate in nursing students, nursing professionals and faculty members should work to understand their gender role stereotypes and patriarchal family environments. Additionally, gender neutrality and integrated gender role identification are required for nursing professionals.”

Reviewer 2 Report

Review of “Do Gender Role Stereotypes of Patriarchal Culture Affect Nursing Students’ Satisfaction?”

This is an interesting and potentially important paper. Worldwide, nursing is moving from being dominated almost totally by women to becoming a profession more men (especially young men) select. The author(s) examine the role that gender stereotypes and patriarchal family environments play in satisfaction with nursing as a major using a sample of nursing students from Korea and standardized, fairly well established measures of concepts and variables. The analysis and conclusions seem fairly sound and valuable.

I have three issues to raise with regard to the analysis and presentation.

First, as a minor modelling issue, there are fairly serious and significant correlations between Patriarchal Family Environment and most of the measures of Gender Role Stereotypes (according to Table 3). Yet, when we move to the regression analysis, PFE is not a statistically significant predictor of MS. It is totally possible that this is actually true, but I wonder if the relationship between PFS and MS is mediated by GRS in the first place. Afterall, the major thing the author(s) argue (implicitly at least) is that patriarchal family structure promotes GRS and that GRS then predicts satisfaction with the major. Like this:

PFS ---------------à GRS ---------------à MS

If this is true, then PFS does affect MS but it does so by affecting GRS – the relationship is mediated or indirect. One could test this by removing GRS from the regression models to see if PFS reaches statistical significance.

Second, it appears that gender is not directly assessed in the regression analysis. Instead, table 1 suggests there are differences in PFS and GRS scores by gender, then men and women are pooled but gender does not appear as a control in the regression analysis. This is fairly important since there are two possibilities ; (1) all of the effect of gender on MS occurs via PFS or GRS, so gender itself would not affect MS net of these controls, or (2) there is still a gender effect net of PFS and GRS that remains unexplained by these variables. Either one of these outcomes is fine and would need to be explained, but in order to do that gender has to be controlled directly in the regression analysis.

Finally, there is a bit of a sample selection issue here that needs to be addressed or at least mentioned. The author(s) are assessing characteristics of students that predict differences in major satisfaction among people who’ve selected the major. The descriptive statistics suggest that there are no differences by gender in major satisfaction. We aren’t sure if there are differences in GRS and PFS by gender because these are not presented (but I suspect there are some).

There are two problems here; (1) men (especially) who had really high scores on GRS or PFS probably wouldn’t enter nursing at all,  and (2) if men and women in nursing systematically differ on GRS and PFS then these could be driving the gender differences in the analysis.

There is no way in your research design to deal with (1) other than to acknowledge it, but it is almost certainly an issue.

For (2) we need to see if there are differences in GRS and PFS by gender (descriptively) and then gender itself has to be added to the MS analysis by itself and then with GSR and PFS added to see if the effects of gender are mediated.

On the other hand, if there are no descriptive differences in GRS or PFS by gender (detected by t-tests), then that itself is very interesting and suggests there are considerable selection effects into nursing that minimize gender differences.

Good luck with your revision and your work.

Author Response

Response to Reviewer 2

This is an interesting and potentially important paper. Worldwide, nursing is moving from being dominated almost totally by women to becoming a profession more men (especially young men) select. The author(s) examine the role that gender stereotypes and patriarchal family environments play in satisfaction with nursing as a major using a sample of nursing students from Korea and standardized, fairly well established measures of concepts and variables. The analysis and conclusions seem fairly sound and valuable.

I have three issues to raise with regard to the analysis and presentation.

  1. First, as a minor modelling issue, there are fairly serious and significant correlations between Patriarchal Family Environment and most of the measures of Gender Role Stereotypes (according to Table 3). Yet, when we move to the regression analysis, PFE is not a statistically significant predictor of MS. It is totally possible that this is actually true, but I wonder if the relationship between PFS and MS is mediated by GRS in the first place. Afterall, the major thing the author(s) argue (implicitly at least) is that patriarchal family structure promotes GRS and that GRS then predicts satisfaction with the major. Like this:

PFS ---------------à GRS ---------------à MS

If this is true, then PFS does affect MS but it does so by affecting GRS – the relationship is mediated or indirect. One could test this by removing GRS from the regression models to see if PFS reaches statistical significance.

Response: We would like to thank you for your constructive suggestions and feedback. They have greatly improved the overall quality of our manuscript.

Accordingly, we tested whether the relationship between patriarchal family environment and MS is mediated by gender role stereotypes by removing gender role stereotypes from the regression models to see if patriarchal family environment reached statistical significance. Ultimately, we found that gender role stereotypes did not demonstrate a mediating effect on patriarchal family environment and MS. We revised as below.

(Page 6, Lines 197­–201)

“However, gender and patriarchal family environment were not statistically significant predictors of MS. Through the extra regression analysis, we found that gender role stereotypes did not play a mediating role in the relationship between patriarchal family environment and MS (B=-0.14, p=.079).”

  1. Second, it appears that gender is not directly assessed in the regression analysis. Instead, table 1 suggests there are differences in PFS and GRS scores by gender, then men and women are pooled but gender does not appear as a control in the regression analysis. This is fairly important since there are two possibilities ; (1) all of the effect of gender on MS occurs via PFS or GRS, so gender itself would not affect MS net of these controls, or (2) there is still a gender effect net of PFS and GRS that remains unexplained by these variables. Either one of these outcomes is fine and would need to be explained, but in order to do that gender has to be controlled directly in the regression analysis.

Response: We would like to thank you for your constructive suggestions and feedback. We agree with your comment. Accordingly, gender was controlled directly in the regression analysis and we revised the manuscript (please see page 6, lines 207–209; Table 4).

  1. Finally, there is a bit of a sample selection issue here that needs to be addressed or at least mentioned. The author(s) are assessing characteristics of students that predict differences in major satisfaction among people who’ve selected the major. The descriptive statistics suggest that there are no differences by gender in major satisfaction. We aren’t sure if there are differences in GRS and PFS by gender because these are not presented (but I suspect there are some).

There are two problems here; (1) men (especially) who had really high scores on GRS or PFS probably wouldn’t enter nursing at all, and (2) if men and women in nursing systematically differ on GRS and PFS then these could be driving the gender differences in the analysis.

There is no way in your research design to deal with (1) other than to acknowledge it, but it is almost certainly an issue.

Response: You are right. We should include this point in the limitation section. Accordingly, we revised it.

(Page 8, Lines 283–287)

“Additionally, the possibility that male students with high gender role stereotypes or patriarchal family environments did not choose nursing as their college major cannot be negated. Subsequent studies would do well to verify the difference in gender role stereotypes and patriarchal family environment between male students who choose nursing majors and male students from other majors.”

For (2) we need to see if there are differences in GRS and PFS by gender (descriptively) and then gender itself has to be added to the MS analysis by itself and then with GSR and PFS added to see if the effects of gender are mediated. On the other hand, if there are no descriptive differences in GRS or PFS by gender (detected by t-tests), then that itself is very interesting and suggests there are considerable selection effects into nursing that minimize gender differences.

Good luck with your revision and your work.

Response: Thank you very much for your helpful insight. As per your direction, we tested whether there were differences in gender role stereotypes and patriarchal family environment by gender and listed the results in Tables 2 and 4. We found that there were differences in gender role stereotypes (t=3.58, p<.001) and patriarchal family environment (t=3.55, p<.001) but not in MS by gender (Table 2); moreover, we also added gender to the MS analysis and found that it was not statistically significant (B=0.06, p=.964, the same result of the t-test). Next, we combined gender with gender role stereotypes and patriarchal family environment to see if the effects of gender were mediated; no significant effect was observed (B=1.55, p=.234). Please see page 6, lines 207–209; Table 4.

Round 2

Reviewer 1 Report

This article reports on an investigation into the relationship between gender role stereotypes, patriarchal family environment, and nursing students satisfaction with their study major. It was conducted in three universities in South Korea.

This is an interesting and contemporary question to investigate, that from a methodological perspective, appears to have been rigorously and well analyzed in this study. I enjoyed my second reading of this article as it has been greatly improved with the authors’ edits. There are still a few things that need attention, as detailed below:

  • There has been an improvement in the use of causal language. However, as this type of study doesn’t justify causal language, there are still a few modifications required. See for example, lines 20, 462, 945, and 948, where ‘effects’ or ‘led to’ should be reframed as relationships, associations, or predictions.
  • In lines 92-96, there appears to be a word or concept missing. Did they perceive the gender roles differently? Of did they perceive the existence of gender role stereotypes in both types of societies? This is not clear.
  • In Table 3, the p- values (currently in brackets) should be removed. It is sufficient to include a key at the bottom of a correlation table, indicating, for example, *** p < 0.001, etc., as you do in Table 4
  • In Table 4, the first column needs to have its titles aligned consistently
  • In several places the authors mention controlling for demographic variables / personal characteristics, but I believe that with the reduced model, this comment is unnecessary.
  • I do not see the relevance of the paragraph in lines 644-657. The variable ‘Year’ has been removed from the final model, so I do not see the connection of this paragraph to the findings from this study.
  • The two paragraphs that go across lines 678-684 need to be linked. For example, start the second paragraph “This is similar to findings from a previous study, that found that…”
  • In lines 938-941 the authors state that “social gender role stereotypes of nursing students can negatively influence MS” when in fact the findings show that (unexpectedly) a rise in social gender role stereotypes is associated with a rise in MS. SO, this relationship is positive, whereas the intellectual gender role stereotypes show a negative association.
  • The end of the discussion section would benefit from a closing, summarizing, paragraph. Something along the lines of “These findings, when taken together, suggest that… (followed by three key takeaways)”
  • In the conclusion section, lines 946 – 951 could benefit from rewriting / restructuring for the sake of clarity. I suggest something like

“This model had an explanatory power of 12.0%, and indicated that being younger in age, having a higher GPA, voluntarily selecting the nursing major, and having lower intellectual gender stereotypes and higher social gender stereotypes were all associated with increased MS. Overall, neither gender stereotypes nor patriarchal cultural background had a negative relationship with the major satisfaction of nursing students in this study."
